# In the Shoulder or in the Brain? Behavioral, Psychosocial and Cognitive Characteristics of Unilateral Chronic Shoulder Pain with Symptoms of Central Sensitization

**DOI:** 10.3390/healthcare10091658

**Published:** 2022-08-30

**Authors:** Paraskevi Bilika, Jo Nijs, Emmanouil Fandridis, Zacharias Dimitriadis, Nikolaos Strimpakos, Eleni Kapreli

**Affiliations:** 1Clinical Exercise Physiology and Rehabilitation Research Laboratory, Physiotherapy Department, University of Thessaly, 35100 Lamia, Greece; 2Pain in Motion Research Group (PAIN), Department of Physiotherapy, Human Physiology and Anatomy, Faculty of Physical Education & Physiotherapy, Vrije Universiteit Brussel, 1050 Brussels, Belgium; 3Chronic Pain Rehabilitation, Department of Physical Medicine and Physiotherapy, University Hospital Brussels, 1050 Brussels, Belgium; 4Hand-Upper Limb-Microsurgery Department, Attika General Hospital KAT, 14561 Kifissia, Greece; 5Health Assessment and Quality of Life Laboratory, Physiotherapy Department, University of Thessaly, 35100 Lamia, Greece

**Keywords:** central sensitization, chronic shoulder pain, catastrophizing, depression, disability

## Abstract

The recognition of central sensitization (CS) is crucial, as it determines the results of rehabilitation. The aim of this study was to examine associations between CS and catastrophizing, functionality, disability, illness perceptions, kinesiophobia, anxiety, and depression in people with chronic shoulder pain (SP). In this cross-sectional study, 64 patients with unilateral chronic SP completed a few questionnaires including the Central Sensitization Inventory, the Oxford Shoulder Score, the Tampa Scale for Kinesiophobia, the Hospital Anxiety and Depression Scale, the Pain Catastrophizing Scale, the Brief Illness Perception Questionnaire and the “arm endurance” test. On the basis of three constructed linear regression models, it was found that pain catastrophizing and depression (model 1: *p* < 0.001, *R* = 0.57, *R*^2^ = 0.33), functionality (model 2: *p* < 0.001, *R* = 0.50, *R*^2^ = 0.25), and helplessness (model 3: *p* < 0.001, *R* = 0.53, *R*^2^ = 0.28) were significant predictors for CS symptoms in chronic SP. Two additional logistic regression models also showed that depression (model 4: *p* < 0.001, Nagelkerke *R*^2^ = 0.43, overall correct prediction 87.5%) and functionality (model 5: *p* < 0.001, Nagelkerke *R*^2^ = 0.26, overall correct prediction 84.4%) can significantly predict the classification of chronic SP as centrally sensitized. Patients who were classified as centrally sensitized (n = 10) were found to have significantly worse functionality, psychological factors (anxiety, depression, kinesiophobia, catastrophizing), and pain intensity (*p* < 0.05). Catastrophizing, depression, and functionality are predictive factors of CS symptoms in patients with chronic shoulder pain. Health care providers should adopt a precision medicine approach during assessment and a holistic rehabilitation of patients with unilateral chronic SP.

## 1. Introduction

In chronic shoulder pain (SP), similarly to other chronic musculoskeletal disorders, several studies have confirmed that the extent of tissue damage may not be related to the occurrence or intensity of SP. Therefore, only a diagnosis of structural pathology in people with SP may not provide adequate information for schema therapy in clinical practice [1,2,3]. 

Modern pain neuroscience has given us new insights into understanding chronic pain, including the role of central sensitization (CS) in the presence and amplification of chronic pain experiences [4,5]. CS is a neurophysiological term that describes the enhancement of neural signaling within the Central Nervous System resulting in hypersensitivity to pain [6]. Pain that can be linked to an altered nociceptive function due to nervous system hypersensitivity has been reported as ‘nociplastic pain’ [7,8,9]. CS has been proven to be evident in a variety of conditions (chronic traumatic neck pain, fibromyalgia, osteoarthritis, migraine, irritable bowel syndrome, chronic fatigue syndrome, and pediatric pain) [10,11,12,13,14,15,16]. Additionally, it is presented in a percentage of patients having low back pain, non-traumatic neck pain, tendinopathies, SP, and tennis elbow [17,18,19,20,21,22]. 

In an effort to understand why only a subgroup of chronic pain patients presents CS, researchers have studied the influence of psychological factors on chronic pain. The revised definition refers that “*Pain is always a personal experience that is influenced to varying degrees by biological, psychological, and social factors*” [23]. Psychological, social, and behavioral factors are clearly associated with treatment outcomes for patients with various musculoskeletal problems [24,25,26]. However, the existence of maladaptive psychosocial and emotional factors is not recommended as a criterion for differentiating chronic pain of CS [8,27]_._ It has been suggested that increased chronic pain and disability are associated with amplified levels of psychosocial distress in people with chronic SP [28], although studies exploring this in patients with chronic SP are lacking and represent an important knowledge gap. On the other hand, some findings support that psychosocial and behavioral factors may enhance central hyperexcitability contributing to strengthening and preserving CS [29]. CS seems to be enhanced by cognitive-emotional states such as catastrophic beliefs, stress, perceived injustice, and anger [29,30,31,32]. 

Psychosocial and cognitive behavioral factors such as incorrect illness perceptions, pain catastrophizing, anxiety, depression, and fear-avoidance play a role in the transition from acute to chronic pain [33,34] and contribute to the extent and severity of pain and disability reported [27] and initiate and robustify the mechanisms of CS without a clear cause-and-effect relationship [35]. Such factors that have been found to have a strong association with CS are increased pain catastrophizing, low sleep efficiency, and increased pain in patients with knee osteoarthritis [36], increased pain, anxiety, depression, poor sleep quality, pain catastrophizing, fear-avoidance in patients with chronic trapezius myalgia [37] and mainly increased pain and pain catastrophizing [38] or individual characteristics involving high trait sensitivity and trait anxiety [39] in patients with chronic low back pain. Although literature offers evidence for the presence of CS in SP, it is rather early to have a clear understanding of issues concerning predisposing factors, mechanisms, and treatment [19]. 

In the current study, the main aim was to examine associations between symptoms of CS and pain catastrophizing, functionality, disability, illness perceptions, kinesiophobia, anxiety, and depression in people with chronic SP. Second, this study aimed to compare the above-mentioned outcome measures between patients presenting and those not presenting symptoms of CS to a clinically relevant degree in a sample of patients having unilateral chronic SP. 

## 2. Materials and Methods

### 2.1. Study Design

The present study is a cross-sectional study concerning patients with chronic shoulder pain who present or not symptoms of CS and follows the STROBE guidelines [40]. The ethics committee of the Physiotherapy Department of the University of Thessaly, Greece (52/16-7-2018) and the KAT University Hospital in Athens (02/15-01-2019) approved the study. Furthermore, the study was registered at clinicaltrials.gov (NCT03838471). Recruitment of participants was done from October 2018 to January 2019.

### 2.2. Participants

A total of 66 participants who suffered from chronic SP were recruited from an orthopedic clinic. Chronic SP was defined as unilateral SP lasting for 3 or more months, with a pain intensity of at least 3 on a 0–10 numerical pain rating scale (on most days of the last 3 months) and the presence of at least 2 positive provocation tests (Hawkins, Jobe, or Neer) [41]. SP conditions included non-specific SP, subacromial pain syndrome, rotator cuff tendinopathy, adhesive capsulitis, acromioclavicular pathology, and/or shoulder osteoarthritis. Diagnosis and physical examination were carried out by a doctor (with 20 years of clinical experience) and the inclusion criteria were assessed through an interview. All participants had an X-ray and an MRI scan. Imaging was used by the doctor to confirm the diagnosis and rule out fractures and pain coming from the neck.

Exclusion criteria included: traumatic musculoskeletal shoulder pain, SP considered to be originated from the cervical region, systemic diseases (rheumatoid arthritis, diabetes, polymyalgia rheumatic), fibromyalgia, neurological dysfunction, cancer, shoulder surgery, participants with SP after fracture, being pregnant or given birth in the preceding year, overconsumption of alcohol or any other recreational drug, an inability to provide informed consent, and/or incomplete questionnaires’ responses. Patients were required to continue with the usual care at least 6 weeks prior to study participation to obtain a steady state. 

The sample of our study was calculated on the basis of the “rule of 10” for regression models [42].

### 2.3. Assessment Tools 

Initially, patients were invited to complete a demographic form. Then, they completed a few questionnaires in random order and performed a functional task. 

The symptoms related to CS were assessed using the Central Sensitization Inventory (CSI-Gr). The CSI contains a “part A” of 25 statements related to current health symptoms, indicative of CS (scored on a 5-point Likert scale ranging from 0–4). A mean score of 40 is the cut-off value for CS [43]. The sample was divided into 2 subgroups, the “CS-group” contained persons with a clinically relevant degree of symptoms of CS (CSI ≥ 40) and the “non-CS-group” contained persons with a lower degree of symptoms of CS (CSI < 40). The CSI has been adapted in Greek and proven excellent reliability (Cronbach’s alpha = 0,996, ICC = 0.991) [44].

Self-reported questionnaires included the Oxford Shoulder Score (OSS), the Tampa Scale for Kinesiophobia (TSK), the Hospital Anxiety and Depression Scale (HADS), the Pain Catastrophizing Scale (PCS) and the Brief Illness Perception Questionnaire (BIPQ). All self-reported questionnaires have been cross-culturally adapted in Greek and used in previous studies [44,45,46,47,48,49].

Self-report functional ability was assessed with the Oxford Shoulder Score in Greek [49]. The Oxford Shoulder Score (OSS) is a 12-item patient-reported outcome specifically designed and developed for assessing the impact on patients’ quality of life of degenerative conditions such as arthritis and rotator cuff problems. Clinical studies have shown that the OSS has high internal consistency and is a valid and reliable measure of patient well-being (ICC = 0.997; SEM = 0.56; SDD = 4%).

The self-reported Tampa Scale for Kinesiophobia is a 17-item questionnaire developed to investigate fear of movement or re-injury [50]. Each question is graded on a four-point Likert scale, with grade 1 representing the “highly disagree” answer and 4 “highly agree”. The score is reversed at 4, 8, 12, and 16. Overall scores range from 17 to 68, with the highest scores outlining the increased fear of movement. An overall score less than or equal to 37 indicates reduced fear of movement. In the current study, the Greek version of the questionnaire was used to assess kinesiophobia, which showed satisfactory reliability (Cronbach’s a = 0.74, ICC = 0.78) [46].

The assessment of anxiety and depression was performed using the Hospital Anxiety and Depression Scale (HADS) in Greek. The HADS scale has been adapted culturally and used by a sample of patients that were treated in a general hospital, as well as in community volunteers [48]. The weighting results showed high internal consistency (Cronbach’s index α = 0.884), a test–retest reliability of 0.944 for the total of the scale, 0.899 for the “Anxiety” factor and 0.837 for the “Depression” factor.

The Brief Illness Perception Questionnaire (B-IPQ) in the Greek version was used to assess patients’ perception of the disease. It studies the cognitive and emotional perceptions of patients about their condition, including consequences, duration of development, self-management, control through treatment, personality, understanding of illness, and worries and emotions on a continuous scale from 0–10 in 8 items. Respondents also had the opportunity to list the 3 most important causal factors in their disease. Its original form has shown good indicators of control-reassurance reliability and predictive validity [51]. The questionnaire has been used in the Greek language for the evaluation of chronic patients [52] and people with rheumatoid arthritis [53]. The score is the sum of all the answers (reverse questions 3, 4, and 7). The higher the score, the greater is the extent to which the patient’s illness perceptions are threatening him or her.

Catastrophic thoughts were estimated with the Pain Catastrophizing Scale [54]. The PCS includes 13 items and are rated on a 5-point Likert scale (0 = never, 4 = always). A score over 30 indicates a clinical degree of pain-related destructive thoughts. The PCS has been adapted in Greek [47]. 

The functional task included the arm endurance test. This test measured the time the patient could hold both arms horizontally out to the side depicting small circles. The test was stopped when the time was more than 90 s or participants’ fingers did not remain above a line level with the elbow when the arm was dependent [55].

To prevent test order effects, the order of questionnaires’ measures was randomized. Furthermore, self-reported measures were completed by using an e-application to collect data optimizing study feasibility, minimizing loss to follow-up, and ensuring data protection. The outcome measures were administered only once.

### 2.4. Statistical Analysis

The normality of data was examined by using the Kolmogorov–Smirnov test. For descriptive characteristics, means (M) and standard deviations (SD) were used as measures of central tendency and dispersion, respectively. Pearson or Spearman correlation coefficients (r_s_) were used for examining correlations between the CSI and the other dependent variables, depending on the normality of the data. Three multiple regression analyses (backward method, entry *p* = 0.05, removal *p* = 0.10) were performed for the prediction of the CSI score. The first multiple regression analysis (model 1) included the predictor variables that were relevant to the psychological condition of the patients (5 predictors: BIPQ, PCS, TSK, HADS_anxiety_, HADS_depression_). The second multiple regression analysis (model 2) included the predictor variables that were relevant to the functional condition of patients (2 predictors: OSS, functional test). The third multiple regression analysis (model 3) was a further analysis of the influence of catastrophizing on CSI by using as predictors its three dimensions (3 predictors: rumination, magnification, helplessness). After the completion of the multiple regression analyses, the CSI was dichotomized on the basis of the known cut-off scores which classified the patients either into a central sensitization group or a non-central sensitization group (CSI ≥ 40 central sensitization and CSI < 40 non-sensitization groups). Then, two logistic regression analyses (backwards method, entry *p* = 0.05, removal *p* = 0.10) were performed for the prediction of patients’ pain mechanism (central sensitization/no central sensitization). The first logistic regression analysis (model 4) included as the predictors variables those relevant to the psychological condition of the patients (5 predictors: BIPQ, PCS, TSK, HADS_anxiety_, HADS_depression_). The second logistic regression analysis (model 5) included as the predictors variables those relevant to the functional condition of patients (2 predictors: OSS, functional test). In this study, it was preferred to construct different models, instead of one that would include all the predictors, as this approach offers the opportunity to understand and discuss better the different roles of functionality and psychology for the prediction of central sensitization. Comparisons of demographic, psychological, functional, and pain characteristics between patients with CS and without central sensitization were performed with independent *t*-tests when data was parametric, Mann-Whitney U-tests when data was non-parametric, and χ^2^ when data was nominal. The significance level was set at *p* = 0.05. All data analyses were performed with the Statistical Package for Social Sciences (SPSS), version 26.0.

## 3. Results

The descriptive characteristics of patients with chronic shoulder pain are presented in detail in Table 1. The variables CSI, HADS_anxiety_, BIPQ, and the functional task presented a normal distribution.

CSI was found to have statistically significant moderate to strong correlations with all the psychological, functional, and pain variables (*r* = 0.34–0.57, *p* < 0.05) (Table 2).

The first multiple regression model was found to significantly fit the data overall (*p* < 0.001, *R* = 0.57, *R*^2^ = 0.33, adj. *R*^2^ = 0.30). During its construction, HADS_anxiety_ (part *r* = 0.01, *p* = 0.92), TSK (part *r* = 0.05, *p* = 0.63), and BIPQ (part *r* = 0.11, *p* = 0.29) were successively removed, whereas only PCS and HADS_depression_ remained as important predictors (Table 3).

The second multiple regression model was also found to significantly fit the data overall (*p* < 0.001, *R* = 0.50, *R*^2^ = 0.25, adj. *R*^2^ = 0.24). During its construction, the functional test (part *r* = −0.12, *p* = 0.27) was removed, whereby only OSS remained as the important predictor (Table 3).

The third multiple regression model was found to significantly fit the data overall (*p* < 0.001, *R* = 0.53, *R*^2^ = 0.28, adj. *R*^2^ = 0.27). During its construction, rumination (part *r* = 0.06, *p* = 0.58) and magnification (part *r* = 0.09, *p* = 0.39) were successively removed, whereby only helplessness remained as the significant predictor (Table 3).

After using logistic regression analysis, model 4 was found to be statistically significant (model 4: χ^2^ (1) = 18.45, *p* = 0.000, Nagelkerke *R*^2^ = 0.43, overall correct prediction 87.5%). During its construction, BIPQ (*p* = 0.98), TSK (*p* = 0.70), PCS (*p* = 0.36) and HADS_anxiety_ (*p* = 0.12) were successively removed, whereby only HADS_depression_ remained as the significant predictor (Table 4).

Model 5 was also found to be statistically significant (model 5: χ^2^ (1) = 10.22, *p* = 0.001, Nagelkerke *R*^2^ = 0.26, overall correct prediction 84.4%). During its construction, the functional task (*p* = 0.46) was removed, and OSS remained as the significant predictor (Table 4).

Patients who were classified with CS were found to have significantly worse psychological, functional, and pain characteristics in comparison with the patients who were classified without CS (*p* < 0.05) (Table 5).

## 4. Discussion 

### 4.1. Main Findings 

This study examined the relationship between symptoms of CS and a battery of psychological and cognitive behavioral factors such as kinesiophobia, anxiety, depression, pain catastrophizing, incorrect illness perceptions, and functionality in patients with unilateral chronic SP. The results revealed that pain catastrophizing, depression, and functionality were strong predictors for CS symptoms in chronic SP (in two different models). When CSI was used to classify the patients either to a central sensitization group or a non-central sensitization group, based on known cut-off scores (CSI ≥ 40 central sensitization and CSI < 40 non-sensitization groups), only depression and functionality remained as strong predictors (in two different models). Furthermore, statistically significant differences were found between tested subgroups with higher levels of maladaptive cognitive-behavioral factors and poorer shoulder function in the CS group. According to the cutoff scores of kinesiophobia and pain catastrophizing, mentioned in the methodology, the CS group had clinically meaningful differences in relation to the non-CS group.

Pain catastrophizing was one important CS predictor found in the current study. Pain catastrophizing is one of the most important features of the Fear Avoidance Model (FAM) which was created 25 years ago to explain the development and persistence of disability presented in a subgroup of chronic low back pain patients [25,56]. This model advocates that pain-related disability is caused by an interacting, recurring sequence of fear-related cognitive, affective, and behavioral processes and it has been used for explaining disability associated with a wide range of musculoskeletal pain conditions [28,35]. 

Following an injury, pain acts as a protective mechanism to prevent further injury. This mechanism includes the reduced functionality of the area and the initiation of cognitive and emotional processes such as fear of movement, which have been established by previous experiences. Probably the context in which the injury takes place but also the presence of psychological distress leads to a vicious circle where destructive thoughts decrease the ability to disengage from pain. Studies in patients with chronic musculoskeletal pain have shown differences in the structure and function of their brain compared to the control group [57]. The mesolimbic and prefrontal areas are mainly associated with such changes. These areas are involved in the processing and cognitive response of incoming stimuli and consequently in the cognitive and behavioral aspects of pain [58]. Disruption of activity in these areas appears to result in increased vigilance and decreased ability to disengage from pain. Interestingly, these areas communicate with areas of the brainstem and sensorimotor centers, ensuring the conditions for causing neuroplastic changes, while also affecting the descending pain modulatory systems [59]. The “top-down” modulation of pain from the brainstem (periaqueductal gray and rostral ventromedial medulla) is an endogenous opioid pain system involving connections to higher brain areas such as the hypothalamus and amygdala. This knowledge provides some explanation as to why pain is not necessarily related to the extent of the injury and at the same time emphasizes that various factors can affect the experience of pain [60].

Although catastrophizing seems to play a role in maintaining or enhancing pain, it is interesting to see which aspect of catastrophizing predicts the symptoms of CS more strongly. In model 3 of the present study, the three aspects of catastrophizing were introduced as predictors (rumination, magnification, helplessness). Only helplessness was an important predictor. Helplessness refers to the feeling of inability to manage pain. In the PCS, items 1, 2, 3, 4, 5, and 12 refer to helplessness and include statements such as “It’s awful and I feel that it overwhelms me” or “There is nothing I can do to reduce the intensity of the pain” [54,61]. Previous studies seem to agree with this finding. A study in patients with chronic low back pain showed that helplessness was more strongly associated with the level of disability [62]. Helplessness was the only factor, out of the three subcategories of catastrophizing, that predicted the severity of the pain and pain interference, while it predicted, along with the “magnification” subscale, the depressive mood and the mental-health-related quality of life [63]. Helplessness encompasses reduced self-efficacy and has been associated with the use of passive pain management strategies. Because CS is associated with poor rehabilitation outcomes, its association with helplessness, in the current study, may be due to failed attempts to treat and alleviate patients with CS [63].

Additionally, the second important CS predictor found in the current study was depression. Pain patients present depression with a mean occurrence rate of approximately 50% [64,65], a percentage that could be regarded as rather extensive, taking into account that depression in the general population exists in less than 10% [66] and the population with major illness does not exceed 20% [67]. Depression is not only associated with the existence of chronic pain but also influences the intensity and duration of pain, being a significant element of pain-related disability and healthcare service utilization [68,69]. Although we are not able to know the cause-and-effect relationship between depression and chronic pain, it is speculated that this is a rather two-way relationship. It seems that the presence of one factor can cause the other to develop or deteriorate. Regarding CS, a study on asymptomatic individuals with high levels of psychological distress promoted the development of chronic widespread pain [70]. In contrast, the presence of widespread pain in people without psychological distress was associated with the future onset of psychological distress [71].

There are a number of proposed mechanisms for deciphering this kind of relationship, such as similar molecular characteristics (e.g., neurotransmitter abnormalities and genetic and epigenetic modifications) [72] or common anatomical structures and regions (e.g., prefrontal cortex, anterior cingulate cortex, thalamus, hippocampus) [73,74]. Relatively, pain processing and mood are known to be controlled by the same neurotransmitters such as serotonin, dopamine, and norepinephrine. Their action through the “top-down” system can achieve the inhibition of pain. In the case of CS pain, there are disorders in this system [75]. At the same time, depression is associated with a reduction in these neurotransmitters [76].

Lastly, the third important CS predictor found in the current study was self-reported shoulder functionality. Difficulty in performing daily activities can be the result of attention to pain or/and an avoidance behavior [56]. Although kinesiophobia did not appear to be a significant predictor of the presence of CS symptoms, in the current study, perhaps catastrophizing was leading to disability. Constant attention to pain, the inability to disengage from pain, and negative thoughts are able to enhance disability. One study in patients with subacute and chronic low back pain showed that there was a link between catastrophizing and disability [77]. Due to the severe symptomatology, increased alertness, decreased parasympathetic function [78], and decreased load tolerance found in CS, it is estimated that disability is increased in this population [79,80]. However, this area is very interesting and more studies are needed, as reduced functionality undermines patients’ physical activity and quality of life [81].

In chronic SP patients, although the existence of psychosocial and cognitive behavioral factors has been reported and their significance in treatment results or even in the post-operative prognosis has been documented [24,82,83,84,85], there is limited evidence of whether these factors are more evident in patients with CS. Our study found that patients with chronic SP and incremental levels of CS presented significantly higher scores in kinesiophobia, anxiety, depression, illness perceptions, and pain catastrophizing and clinically meaningful differences in pain catastrophizing and kinesiophobia than patients with depleted levels of CS. More interestingly, the symptoms of CS were significantly correlated with all examined psychosocial and cognitive behavioral factors, with pain catastrophizing and depression being verified as considerable explanatory factors for CS symptoms in chronic SP patients according to the regression analysis. Furthermore, our results showed that function was significantly more limited with increased pain in chronic SP patients with CS in comparison with non-CS patients. Likewise, function (examined by OSS and task) and pain were significantly correlated with CS symptoms. Our results are in accordance with previous studies examining the association of psychosocial and cognitive behavioral factors with CS [36,37,38,39,86]. 

Following previous studies, it seems that there is a subgroup of patients with unilateral chronic SP that presents incremental levels of CS symptoms. In the current study, this subgroup represented 15,6% of the sample. Compared to the non-CS group, these patients did not differ in age or gender; however, they presented higher pain, disability, and cognitive behavioral states. Similar subgroups have been identified in some other chronic musculoskeletal disorders such as in patients with low back pain [87], non-traumatic neck pain [17], tendinopathies [88], rheumatoid arthritis [89], pain in cancer survivors [90], and tennis elbow [20]. On the contrary, CS seems to be a predominant feature in patients with chronic traumatic neck pain (i.e., whiplash) [31], fibromyalgia [86], osteoarthritis [11], migraine [16], irritable bowel syndrome [13], chronic fatigue syndrome [12], and pediatric pain [15].

The results of the present study support the biopsychosocial framework within which clinicians should assess and treat chronic SP patients. There is strong evidence that chronic pain patients should be treated holistically, in view of their psychosocial, and cognitive behavioral states. These factors influence their symptoms (pain and function), and they are considered detrimental to the success or failure of any therapeutic schema. A lot of patients are undergoing surgery to manage chronic SP, even without any substantial tissue damage, to find a solution to their extensive chronic pain and disability even in daily activities. However, in a subgroup of patients, surgery or drugs not only failed to provide positive effects but also aggravated symptoms [82,91]. In fact, there was a rapid increase in the rate of dependence on prescription opioids for the treatment of chronic pain. Now, It seems that psychological factors played a role in this complication as well [92]. The treatment of patients having CS symptoms should not focus only on targeting local anatomical structures, but instead factors that sustain the process of CS should be recognized and manipulated. There is strong evidence that a “issues in the tissues but pain in the brain” pattern exists in patients with increased CS symptoms and new intervention concepts should be addressed in health care professionals. 

### 4.2. Limitations

Although the sample size seems to be sufficient for the examination of correlations and the construction of regression models, the examination of differences between the CS and the non-CS group was not supported by a sample size calculation. Furthermore, the CSI contains questions about symptoms and signs related to CS; however, it is not adequate by itself to provide a diagnosis of CS [4]. Therefore, the CSI was used in the current study in order to differentiate patients presenting increased symptoms of CS (≥40 score) and decreased symptoms of CS (<40 score) [93], in accordance with previous research [38,43]. Quantitative Sensory Testing is used as a measure of CS and may be used in future studies [94]. Furthermore, criteria for identifying patients with nociplastic pain have recently been published that could be used in future studies. However, the testing of their psychometric properties is required [8]. The functional test we selected was a non-overhead task, which has been used in patients with chronic pain and has shown excellent inter-rater reliability, moderate test–retest reliability, and face validity. However, a group of patients (such as patients with adhesive capsulitis) could not perform the test and this is a limitation. In future studies, it would be beneficial to use other functional tests to compare results with previous studies.

### 4.3. Implications for Rehabilitation

It seems that there is an important subgroup of patients (15.6%) with unilateral chronic SP that presents incremental levels of CS symptoms. Τhe presence of CS has been associated with a poor outcome when used classical treatments (surgery, electrotherapy, etc.). The results of the present study support the biopsychosocial (lifestyle) framework within which clinicians should assess and treat chronic SP patients. Factors such as catastrophizing, depression, and functionality may play an important role in the success or failure of any therapeutic schema. The findings of our study are in agreement with other studies of patients with chronic low back pain [38] and knee pain [95,96]. Therefore, healthcare professionals should develop the appropriate skills to manage patients with chronic pain. There is a need to move towards precision pain medicine in which patients will be classified according to characteristics related to evaluation (biological, psychological factors, and others), prognosis and patient characteristics in order to provide patients with the appropriate personalized treatment.

## 5. Conclusions

Symptoms of CS were significantly associated with psychosocial and cognitive behavioral factors whereas pain catastrophizing and depression seemed to be the most influencing factors in patients with unilateral chronic SP. Patients with unilateral chronic SP and symptoms of CS presented higher scores in kinesiophobia, anxiety, depression, illness perceptions, and pain catastrophizing, and lower scores in shoulder function. From the above results, unilateral chronic SP should not be considered as a musculoskeletal problem only, but clinicians should adopt a precision medicine approach during assessment and a holistic rehabilitation of patients with unilateral chronic SP.

## Figures and Tables

**Table 1 healthcare-10-01658-t001:** Descriptive statistics of the study (*N* = 64).

	Parameters	Mean ± SD
Variables	Age (years)	52.02 ± 13.12
	Pain_now	5.56 ± 1.6
	Pain_usually	5.86 ± 1.5
	CSI	26.13 ± 11.01
	PCS	20.14 ± 11.78
	HADS_anxiety_	6.44 ± 4.08
	HADS_depression_	6.42 ± 4.21
	TSK	38.09 ± 10.34
	BIPQ	38.95 ± 11.40
	OSS	38.05 ± 11.21
	Functional Task	24.58 ± 9.84
		**n(%)**
Gender	Female	33 (51.6)
	Male	31 (48.4)
Education	Primary Education	11 (17.2)
	Secondary Education (Gymnasio)	6 (9.4)
	Secondary Education (Lykeion)	15 (23.4)
	Tertiary Education	32 (50.0)
Language	Greek	64 (100)
Place of Residence	Athens, Greece	64 (100)

CSI = Central Sensitization Inventory, PCS = Pain Catastrophizing Scale, HADS = Hospital Anxiety and Depression Scale, TSK = Tampa Scale for Kinesiophobia, BIPQ = Brief Illness Perception Questionnaire, OSS = Oxford Shoulder Scale.

**Table 2 healthcare-10-01658-t002:** Correlations between CS symptoms and psychosocial, emotional, and functional parameters (*N* = 64).

Variables	*r*	*p*
Pain Now	0.48	<0.001
Pain Usually	0.49	<0.001
PCS	0.57	<0.001
BIPQ	0.49	0.001
TSK	0.34	0.006
OSS	0.50	0.001
HADS Anxiety	0.42	0.001
HADS Depression	0.48	0.001
Functional Task (sec)	−0.44	0.001

PCS = Pain Catastrophizing Scale, HADS = Hospital Anxiety and Depression Scale, TSK = Tampa Scale for Kinesiophobia, BIPQ = Brief Illness Perception Questionnaire, OSS = Oxford Shoulder Scale.

**Table 3 healthcare-10-01658-t003:** Multiple regression analysis for the prediction of CSI scores from psychological and functional variables and the different dimensions of catastrophizing (*N* = 64).

	Variable	Part *r*	B	SE B	*β*	*p*
Model 1	Constant		14.70	2.40		0.000
	PCS	0.32	0.37	0.12	0.40	0.004
	HADS Depression	0.19	0.62	0.35	0.24	0.077
Model 2	Constant		16.74	9.34		0.078
	OSS	0.50	0.49	0.11	0.50	0.000
Model 3	Constant		17.66	2.07	0.53	0.000
	PCS Helplessness	0.53	1.02	0.21	0.53	0.000

OSS = Oxford Shoulder Score, HADS = Hospital Anxiety and Depression Scale, PCS = Pain Catastrophizing Scale, B = unstandardized beta, SE B = standard error of the unstandardized beta, *β* = standardized beta.

**Table 4 healthcare-10-01658-t004:** Logistic regression analysis for the prediction of CSI classification (central or non-central sensitization) from the different psychological states and the functional variables of patients with chronic shoulder pain (*N* = 64).

	Variable	B	SE B	Exp(B)	95% CI of Exp(B)	*p*
Model 4	Constant	−5.78	1.57	0.00		0.000
	HADS Depression	0.47	0.15	1.60	1.19–2.15	0.002
Model 5	Constant	−6.75		0.00		0.002
	OSS	0.12	0.05	1.13	1.03–1.23	0.009

HADS = Hospital Anxiety and Depression Scale, OSS = Oxford Shoulder Score, B = unstandardized beta, SE B = standard error of the unstandardized beta, Exp(B) = exponential value of B, 95% CI of Exp(B) = 95% confidence intervals of the exponential value of B.

**Table 5 healthcare-10-01658-t005:** Descriptive statistics and statistical analysis of groups with and without central sensitization symptoms.

Variables	Non CS-Group (*n* = 54)	CS-Group (*n* = 10)	Tests	*p*	95% CI	
	Mean ± SD				Lower	Upper
Age	51.06 ± 12.93	57.20 ± 13.59	Independent *t*-test	0.210	−16.24	3.95
Gender (Male n(%))	25 (46.3%)	6 (60%)	×2	0.426		
Pain Now (0–10)	5.35 ± 1.58	5.69 ± 1.16	Mann-Whitney U-test	0.008	0.006	0.010
Pain Usually (0–10)	6.70 ± 1.48	6.80 ± 1.32	Mann-Whitney U-test	0.031	0.027	0.033
CSI	23.15 ± 9.24	42.20 ± 2.3	Independent *t*-test	0.000	−21.96	−16.14
PCS	17.96 ± 11.07	31.90 ± 8.28	Mann-Whitney U-test	0.000	0.000	0.009
TSK	36.76 ± 10.36	45.30 ± 7.04	Mann-Whitney U-test	0.011	0.009	0.013
BIPQ	36.83 ± 12.45	50.40 ± 13.13	Independent *t*-test	0.010	−23.32	−3.82
HADS Anxiety	5.56 ± 3.60	11.20 ± 3.19	Independent *t*-test	0.000	−8.06	−3.23
HADS Depression	5.52 ± 3.82	11.30 ± 2.67	Mann-Whitney U-test	0.000	0.000	0.000
OSS	36.22 ± 10.83	47.90 ± 7.89	Mann-Whitney U-test	0.000	0.000	0.005
Functional Task (sec)	25.99 ± 9.57	16.99 ± 7.91	Independent *t*-test	0.006	2.97	15.03

CSI = Central Sensitization Inventory, OSS = Oxford Shoulder Score, TSK = Tampa Scale for Kinesiophobia, HADS = Hospital Anxiety and Depression Scale, PCS = Pain Catastrophizing Scale, BIPQ = Brief Illness Perception Questionnaire.

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
