# Peer review of "In the Shoulder or in the Brain? Behavioral, Psychosocial and Cognitive Characteristics of Unilateral Chronic Shoulder Pain with Symptoms of Central Sensitization"

_healthcare, 2022, doi:10.3390/healthcare10091658_

Round 1

Reviewer 1 Report

Dear authors,

this is a nice and interesting project and it strengthens the biopsychosocial approach to healthcare in shoulder pain patients, which is utterly important. 

A few questions remain open.

Thank you for considering them.

Reviewer 2 Report

Thank you for inviting me to review the manuscript ‘In the Shoulder or in the Brain? Behavioral, Psychosocial and 2 Cognitive Characteristics of Unilateral Chronic Shoulder Pain 3 with Symptoms of Central Sensitization’.

Methods

·       Section2.2: Your exclusion criteria are extensive leaving a specific population of people. However, we know very little about them (Table 1). If we are to truly understand this population, we should know more about them as referred to in your introduction. Do you have any further demographics? If not, can you justify why you left out such criteria as defined by Cochrane Progress plus criteria [https://methods.cochrane.org/equity/projects/evidence-equity/progress-plus]?

·       Section 2.3: Please justify why you chose this functional arm test? What was its purpose? It would be biased to certain pathologies.

·       You describe your study as observational but then chose to compare groups without reference to a power calculation. Please comment on this in the methods and limitations.

·       Table 5: Please provide the 95% CI for the differences between the groups for reader clarity.

·       Table 5: Male % should be 60% not 6%.

·       I think you have identified important clinically relevant characteristics, i.e., catastrophising and depression. Please highlight 15% of the sample in the implications for rehabilitation as this is not an insignificant number of the population. How does it compare to other studies?

·       Section 4.1: You say “Furthermore, statistically significant differences were found between tested subgroups with higher levels of maladaptive cognitive behavioral factors and poorer shoulder function in the CS group.” Were they clinically meaningful differences?

Reviewer 3 Report

-    The present manuscript deals with a very interesting and complex issue of the central sensitization and its association with the chronic shoulder pain psychosocial and cognitive behavioral factors, such as pain catastrophizing, functionality, disability, illness perceptions, kinesio-phobia, anxiety and depression.

-    Overall, this research should be considered as a very good quality piece of work. It presents good methods, robust statistics, and the discussion of the results is supported by neurophysiological explanations.

-    A very interesting finding and a strength of this study is the analysis of the predictor factors of the CS associated with the chronic pain. This could be better introduced in the title of this manuscript.

-    The layout of the manuscript has a well-developed schema and guides the reader to the specific parts of the study

-    The background knowledge is well presented however, a clear definition of the central sensitization in the first paragraph of the second page (line 46-52) would help reader to make the connection with the “nociplastic pain”.

-    The abbreviations should be used in a better and the same way throughout the manuscript. (i.e. the chronic shoulder syndrome is referred to abbreviation ‘CSP’ in the abstract section but as “chronic SP” in the rest of the text. This should be written in the same way in the whole document)

-    Sources within the text should follow the same format (i.e. Vancouver). At lines 66, 131, 143, 147 these are inserted in Harvard system.. correct these.

-    Grammar errors are widespread through the text. Please check the whole document and correct (i.e. Section 2.3, line 117, grammar correction “The symptoms…were (not was) assessed…”, Section 2.3, Line 167: grammar error “…or participants’ fingers weren’t (not wasn’t) remained…”

-    Section 2.3, lines 118-164 presents the assessment tools and not the procedure. As such, this should be considered as a separate subsection in the materials section (and not under the title “procedure”)
